# Integrative Taxonomy Revealed High Diversity of *Hemiphyllodactylus* Bleeker, 1860 (Squamata: Gekkonidae) and the Description of Three New Species from Yunnan Province, China

**DOI:** 10.3390/ani14203030

**Published:** 2024-10-19

**Authors:** Hongxin Zhou, Xiuyan Li, Chaoying Yuan, Liangwei Cui, Shuo Liu, Dingqi Rao

**Affiliations:** 1Kunming Institute of Zoology, Chinese Academy of Sciences, Kunming 650201, China; 15758797016@163.com (H.Z.);; 2Key Laboratory for Forest Resources Conservation and Utilization in the Southwest Mountains of China, Ministry of Education Faculty of Biodiversity and Conservation, Southwest Forestry University, Kunming 650224, China; cuilw@eastern-himalaya.cn; 3School of Ecology, Hainan University, Haikou 570228, China; 4Kunming Natural History Museum of Zoology, Kunming Institute of Zoology, Chinese Academy of Sciences, Kunming 650223, China

**Keywords:** cryptic species, discovery, karst, slender geckos, South China

## Abstract

The *Hemiphyllodactylus* is a small gecko. Recently, many species belonging to the genus *Hemiphyllodactylus* have been discovered in the karst region. Therefore, we conducted a large scale of surveys and specimen collections in multiple karst areas of Yunnan Province. Three populations of *Hemiphyllodactylus* were found to be morphologically and molecularly distinct from known species, prompting us to describe them as three new species. Our study further illustrates the high diversity of *Hemiphyllodactylus* species within Yunnan Province.

## 1. Introduction

The karst landform is a diverse landscape formed from rocks that are soluble in water and is a result of the combined effects of mechanical and chemical weathering [1]. This type of terrain hosts highly diverse and localized flora, leading to the formation of unique karst forest landscapes [2]. These diverse landscapes also encompass various microhabitats for resident organisms, fostering increased potential for the evolution of numerous species within the karst landforms [2,3,4,5,6].

Recently, many species belonging to the genus *Hemiphyllodactylus* have been discovered in karst regions [7,8,9]. They have a small body size (SVL < 6 cm), cryptic coloration, low densities [9], and limited dispersal ability [8,10]. They are distributed in South Asia, Southeast Asia, and the Indo–Pacific region [10,11] and currently comprise a total of 59 species [12].

The Yunnan Province in China has extensive karst landforms and is also the richest area for *Hemiphyllodactylus* taxa [9,12]. Grismer et al. (2018) [7] conducted a two-week reptile survey in Myanmar, identifying 12 new gecko species and indicating the presence of numerous undescribed species in karst landscapes [7]. Given the geographic proximity and similar climate characteristics between Yunnan Province in China and Myanmar, similar patterns may exist [8,9,13].

Recent genus-wide molecular phylogenetic studies have indicated the presence of five clades of *Hemiphyllodactylus* within China [7,8,9,11,13,14,15,16,17,18,19,20,21]. Apart from the uncertain location of *H. typus* in China [8], the remaining four clades are distributed in Yunnan Province, specifically clade 3, clade 4, clade 6, and clade 7 as defined by Agung et al. (2022) [9]. This suggests the presence of at least four colonization routes into Yunnan Province [9]. Biogeographical research on *Hemiphyllodactylus* has previously been confined to specific sites [16,17], and there has been no publication detailing the colonization routes for the entire genus. This complexity can be attributed to the topographical intricacies and geographic events involving the Chinese and Indian plates, compounded by a lack of foundational data and, thus, necessitating targeted surveys of *Hemiphyllodactylus*.

Previously, all *Hemiphyllodactylus* populations within Yunnan were classified as *Hemiphyllodactylus yunnanensis* Boulenger, 1903. It was not until Grismer et al. (2013) [11] integrated molecular data into the taxonomic study of *Hemiphyllodactylus*, elevating the three subspecies of *H. yunnanensis* to full species. Consequently, the *Hemiphyllodactylus* species within Yunnan Province have been gradually described. Currently, there are nine species of *Hemiphyllodactylus* in Yunnan Province: *H. yunnanensis*; *H. jinpingensis* Zhou and Liu, 1981; *H. dushanensis* Zhou and Liu, 1981; *H. longlingensis* Zhou and Liu, 1981; *H. changningensis* Guo, Zhou, Yan and Li, 2015; *H. zhutangxiangensis* Agung, Grismer, Grismer, Quah, Chornelia, Lu and Hughes, 2021; *H. simaoensis* Agung, Chornelia, Grismer, Grismer, Quah, Lu, Tomlinson and Hughes, 2022; *H. yanshanensis* Agung, Chornelia, Grismer, Grismer, Quah, Lu, Tomlinson and Hughes, 2022 and *H. gengmaensis* Zhou, Li, Liu and Rao, 2023. However, populations found in other locations within this area may represent distinct species.

An extensive survey was conducted in Yunnan Province, China, known for its richness in *Hemiphyllodactylus* species. The survey has provided significant foundational data for the taxonomic and biogeographical study of *Hemiphyllodactylus*. Consequently, this study has led to the initial discovery of some new species. Based on the phylogenetic relationships and morphological characteristics, we have described three new species and supplemented new distribution locations and molecular sequences for certain known species.

## 2. Materials and Methods

### 2.1. Specimen Collection

Between 2012 and 2023, we conducted multiple surveys of karst landforms in various regions of Yunnan Province, China (Figure 1). We manually collected 114 specimens of *Hemiphyllodactylus* from 17 locations (Table 1) during the active period of *Hemiphyllodactylus* species, which is from 19:00 to 00:00 at night. We fill a sealable wide-mouthed bottle with three quarters of water, then add 5–10 drops of clove oil. After placing the live specimen into the bottle, we tightly screw the lid to euthanize the specimen. Then, we took samples of muscle or liver tissue from each individual specimen and preserved them in 95% ethanol for genetic analysis. The specimens were then stored in 80% ethanol for morphological measurements. All specimens were deposited in Kunming Institute of Zoology (KIZ), Chinese Academy of Sciences (CAS).

### 2.2. DNA Extraction, Amplification, and Sequencing

Trelief Hi–Pure Animal Genomic DNA Kit was used to genomic DNA extraction following the manufacturer’s protocol (www.tsingke.com.cn, accessed on 15 January 2024). We amplified and sequenced the complete mitochondrial NADH dehydrogenase subunit 2 gene (ND2), totaling 1144 bp used the primers L4437b and H5934 [22]. The protocol for polymerase chain reaction (PCR) amplifications followed Agung et al. (2021) [8]. Genomic DNA extraction, PCR processes, and sequencing were executed at Beijing Tsingke Biotechnology Co., Ltd. (Beijing, China) All new sequences have been deposited in GenBank (Appendix A).

### 2.3. Molecular Data and Phylogenetic Analyses

We obtained 1038 bp of NADH dehydrogenase subunit 2 (ND2) sequence data from 296 specimens from GenBank [8,9,23,24,25,26,27,28], containing 292 sequences of the extant *Hemiphyllodactylus* species and 4 ND2 sequences of outgroup taxa: *Gehyra felmani* (Taylor), *G. mutilata* (Wiegmann), *Hemidactylus frenatus* Duméril and Bibron, and *Lepidodactylus lugubris* Duméril and Bibron were used to root the tree based on Heinicke et al. (2011) [29]. These with our 114 new sequences (Appendix A) were used for phylogenetic analyses. Sequences were assembled and manually proofread in SeqMan (DNASTAR, Inc., Madison, WI, USA), then aligned using Clustal W [30] implemented in MEGA 7 [31]. After alignment, we used Gblock 0.91b [32] to remove misaligned positions. ModelFinder v. 2.2.0 [33] was used to select the best-fitting model of evolution based on the Bayesian Information Criterion (BIC). Maximum likelihood (ML) analysis was used to TVM+F+I+G4 as the best fit substitution model for codon position one, TPM3u+F+G4 for position two, and TIM+F+G4 for position three. For phylogenetic relationships analysis, we used maximum likelihood (ML) and Bayesian inference (BI) by IQ–TREE v. 2.2.0 [26] and MrBayes v. 3.2.7a [34] in Phylosuite application [35,36], respectively. We applied 1000 bootstrap pseudoreplicates with the ultrafast bootstrap approximation algorithm (UFBoot), where nodes having values 95 and above were considered highly supported [37]. Bayesian inference (BI) analysis was used to GTR+I+G+F as the best fit substitution model. We ran two independent Markov Chain Monte Carlo (MCMC) analyses with four chains in each analysis (three hot and one cold). We ran the MCMC analyses for 50 million generations, sampled every 50,000 generations, and discarded the first 10% of each run as burn–in. Interactive Tree Of Life (ITOL) was used to draw the phylogenetic tree. MEGA 7 [31] was used to calculate uncorrected pairwise divergence (based on the Kimura 2-parameter) among and within the related species.

### 2.4. Morphological Measurements and Analyses

Mensural data were taken with digital calipers to the nearest 0.01 mm under a dissecting microscope (Jiangnan XTB–01) following Zug (2010), Grismer et al. (2013), and Agung et al. (2021) [8,10,11]: snout–vent length (SVL); tail length (TailL); trunk length (TrunkL); head length (HL); head width (HW); eye diameter (ED); snout–eye length (SnEye); nares–eye length (NarEye); and snout width (SnW).

For meristic characters, we counted the following: chin scales (Chin scale); circumnasal scales (CN); internasals (IS), supralabial scales (SL) and infralabial scales (IL); ventral scales (VS); dorsal scales (DS); lamellar formula, determined as the number of U–shaped subdigital lamellae on the digital pads on digits 2–5 of the hands and feet; the number of subdigital lamellae wider than long on the first finger (SL1F) and toe (SL1T); precloacal and femoral pore series separate or continuous, the total number of precloacal and femoral pores in males; and the number of cloacal spurs on each side.

For coloration characters: we evaluated the following: presence or absence of a dark postorbital stripe extending to at least the neck; presence or absence of dark dorsolateral light-colored spots on the trunk; presence or absence of dark dorsolateral and ventrolateral stripe; presence or absence dark dorsal transverse blotches; presence or absence of dark reticulate pattern on dorsum and the presence or absence of anteriorly projecting arms of the postsacral marking.

We compared the morphological characteristics of the newly constructed operational taxonomic units (OTU) from the phylogenetic analysis with related species that have already been published (Appendix A) in order to determine morphological differences. For statistical analyses, we follow Luu et al. (2023) [24]. All morphological analyses were conducted in R v.4.3.1 [38]. To avoid the potential impact of differential growth rates between male and female individuals on the data analyses, we utilized the GroupStruct package in R to adjust the measurement characteristics of the species used for analysis prior to conducting the analysis: Xadj = log(X) − β[log(SVL) − log(SVLmean)], where Xadj = adjusted value; X = measured value; β = unstandardized regression coefficient for each population; and SVLmean = overall average SVL of all populations [39,40,41,42]. Due to a large amount of missing data for tail length (breakage, damage, regeneration), Tail length was was excluded from the statistical analyses [8,9,24]. Additionally, the extended specimen preservation time resulted in missing measurements for some specimens’ subdigital lamellae, and precloacal and femoral pores were only present in males, hence SL1F, SL1T, the number of U-shaped subdigital lamellae on digits II–V of the hands and feet, and the total number of femoral and precloacal pores were not incorporated in the analysis. After correcting the measured values, to avoid potential confusion between intraspecific and interspecific variation, we normalized the morphological measurement data for each species and compiled them into a single dataset [43,44]. We conducted Levene’s test on the adjusted measurement and meristic features to identify features with statistically similar variances (*p* > 0.05). Subsequently, we performed ANOVA analysis on features with statistically similar variances to determine if there were statistically significant mean differences in the dataset (*p* < 0.05). For features with statistically significant differences, Tukey’s Honestly Significant Difference (TukeyHSD) test was carried out to identify which groups exhibited significant differences.

For principal component analysis and discriminant analysis of principal components, we followed the methods outlined by Luu et al. (2023) [24]. For the spatial relationships of morphological traits, we employed Principal Component Analysis (PCA) based on the corrected measurement values and meristic features in the ADEGENET package in R [45] to determine their position in relation to assumed species boundaries defined by the molecular phylogenetic analysis and univariate analyses (as described above). PCA, implemented using the “prcomp()” command in R, is an indiscriminate analysis plotting the overall variation among individuals (i.e., data points) while treating each individual independently (i.e., not coercing data points into pre–defined groups). Following PCA, Discriminant Analysis of Principal Components (DAPC) was used to test for morphological spatial differences assumed between species. DAPC a priori groups the individuals of each predefined population inferred from the phylogeny into separate clusters (i.e., plots of points) bearing the smallest within-group variance that produce linear combinations of centroids having the greatest between-group variance [45]. DAPC relies on standardized data from its own PCA as a prior step to ensure that variables analyzed are not correlated and number fewer than the sample size. We employed 90–95% of the principal components in the dataset for DAPC analysis [24,45]. To enhance interpretability, we visualized each step in R.

### 2.5. Species Delimitation

Our species definition and species delimitation methods are based on Agung et al. (2022) [9]. First, we hypothesize potential new species by marking those lineages in the phylogenetic topology that did not cluster within the existing named species. Second, we measured the uncorrected pairwise genetic distances between these novel lineages and either known species or other putative species lineages, with a 3.0% difference in mtDNA ND2 considered minimal to define a potential new species. Third, for those lineages exhibiting a genetic difference exceeding 3.0% from their nearest relatives, we conducted a detailed examination of their morphological distinctiveness compared to closely related species. If all three criteria were met, the lineage was considered a confirmed new species.

## 3. Results

### 3.1. Phylogenetic Analyses

Our phylogenetic results are consistent with the entire genus phylogeny tree constructed by Agung et al. (2021) [8], and we named all clades following Grismer et al. (2020) [18] (Figure 2).

All the 114 newly collected samples from this work are divided into clade 3, clade 4, and clade 6 of Agung et al. (2021) [8]. All specimens were nested and formed OTUs as follows: 1 specimen (201600686) from Donghe Township, Lancang County in clustering with *Hemiphyllodactylus zhutangxianensis*; 2 specimens from Tengchong County in clustering with *H. longlingensis*; 14 specimens from Yunlong County, 1 specimen from Yangbi County, and 1 specimen from Yongde County in clustering with *H. changningensis*; 3 specimens from Lvchun County, 3 specimens from Dadugang Township, Jinghong City, 1 specimen from Shangyong Village, Mengla County, and 1 specimen from Nannuoshan Township, Menghai County in clustering with *H. simaoensis*; 18 specimens from Lvchun County, 1 specimen from Nanhua County, 6 specimens from Ziwu Village, Chuxiong City, 4 specimens from Shang Village, Yimen County, and 14 specimens from Xinjie Town, Yuanyang County in clustering with *H. jinpingensis*; 7 specimens from Nanhua County, and 3 specimens from Chuxiong City in clustering with the *H. yunnanensis* complex; and 1 specimen (057601) of unknown origin is embedded within clade 6 of Agung et al. (2022) and is recognized as a new OTU. In total, 9 specimens from Langdao Township, Menglian County form a newly discovered OTU in clade 4; 24 specimens from Mengsong Village are divided into 2 OTUs, where 13 specimens (Mengsongcun A) belong to clade 3, and the remaining 11 specimens (Mengsongcun B) belong to clade 4. We will describe these three new species in the following text.

### 3.2. Genetic Distance

Uncorrected genetic *p*-distances within clades 3 and 4 are provided in Table 2 and Table 3, respectively. The uncorrected genetic *p*-distances of *Hemiphyllodactylus menglianensis* sp. nov. is 5.7% *(H. menglianensis* sp. nov. versus *H. simaoensis*) to 11.1% (*H. menglianensis* sp. nov. versus *H. chiangmaiensis*). Uncorrected genetic *p*-distances for the 13 specimens from Mengsong Village ranges from 5.2% (*H. mengsongcun* sp. nov. versus *H. simaoensis*) to 16.4% (*H. mengsongcun* sp. nov. versus *H. chiangmaiensis*), with their sister group being *H. simaoensis*. Uncorrected genetic *p*-distances for the 12 specimens from Mengsong Village spans from 8.5% (*H. jinghongensis* sp. nov. versus *H. zhutangxiangensis*) to 35.8% (*H. jinghongensis* sp. nov. versus *H. flaviventris*). These uncorrected genetic *p*-distances are all greater than 3%, which was specified as the minimum genetic *p*-distance for species delineation in the previous text.

### 3.3. Morphological Analysis of Menglian and Mengsong B Population

We conducted an analysis using the morphological characteristics of *Hemiphyllodactylus jinpingensis* and *H. simaoensis*. The uncorrected genetic *p*-distances between them are shown in Table 3. Levene’s test results indicated that SVL, HL, SnEye, Chin, IS, SL, IL, VS, and DS have statistically similar variances. The ANOVA analysis results revealed statistically significant average differences for SVL, HL, SnEye, Chin, SL, IL, VS, and DS. The results of the TukeyHSD test and ANOVA analysis were presented in Table 4. Variation in all morphometric and metric characters are visualized in Figure 3.

The Principal Component Analysis has demonstrated that PC1 and PC2 together accounted for 70.50% of the variability, as shown in Figure 4. PC1 represented 49.70% of the dataset variability (Appendix A), with the heaviest loadings for DS, VS, and Chin, while PC2 represented 20.80% of the dataset variability, with the heaviest loadings for Chin and HL. The PCA (Figure 4) revealed substantial overlap between *H. jinpingensis* and *H. simaoensis*, similar to the results of Agung et al. (2022). *Hemiphyllodactylus mengsongcunensis* sp. nov. exhibited slight overlap with *H. simaoensis*, *H. jinpingensis*, and *H. menglianensis* sp. nov. *Hemiphyllodactylus menglianensis* sp. nov. was well–separated from the other species, with only slight overlap with *H. mengsongcunensis* sp. nov. The DAPC results demonstrated well–defined separation of the populations. *Hemiphyllodactylus mengsongcunensis* sp. nov. exhibited minimal overlap with *H. jinpingensis* and *H. simaoensis*, while *H. menglianensis* sp. nov. exhibited no overlap with the other three species. Additionally, the confidence ellipses for *H. menglianensis* sp. nov. and *H. mengsongcunensis* sp. nov. had no overlap with the other species (Figure 4).

### 3.4. Morphological Analysis of Mengsong a Population

We conducted an analysis using the morphological characteristics of *Hemiphyllodactylus zhutangxiangensis*, *H. gengmaensis*, and *H. changningensis*. Levene’s test results showed that SVL, TrunkL, HL, ED, SnEye, NarEye, SnW, Chin, IS, SL, IL, VS, and DS exhibited statistically similar variances. The ANOVA analysis revealed statistically significant average differences for TrunkL, HL, ED, SnEye, SnW, Chin, VS, and DS. The results of the TukeyHSD test and ANOVA analysis were presented in Table 5. Variation in all morphometric and metric characters are visualized in Figure 5.

The Principal Component Analysis has demonstrated that PC1 and PC2 accounted for 63.80% of the variability (Appendix A), as depicted in Figure 6. PC1 represented 47.80% of the dataset variability, with the heaviest loadings for DS, VS, and CN, while PC2 represented 16.00% of the dataset variability, with the heaviest loadings for VS, DS, CN, SL, and Chin. The PCA revealed (Figure 6) that *H. changningensis* overlapped with *H. zhutangxiangensis*, while *H. jinghongensis* sp. nov. overlapped with *H. zhutangxiangensis* and *H. gengmaensis*, and *H. gengmaensis* displayed slight overlap with *H. jinghongensis* sp. nov. based on the DAPC results utilizing the first four principal components, the four species were well separated, and their confidence ellipses showed no overlap (Figure 6).

Based on the phylogenetic relationships, uncorrected genetic *p*-distances, and morphological spatial relationships, we assert that *Hemiphyllodactylus* specimens from Langdao Township, Menglian County, and Mengsong Village in Jinghong City have evolved into new species. Therefore, we will proceed to describe them as three new species below.

## 4. Taxonmy

### 4.1. Hemiphyllodactylus menglianensis sp. nov. (Figure 7)

http://zoobank.org/urn:lsid:zoobank.org:act:D3306BE7-0B70-4E94-9A02-7D6123C0CCAE (accessed on 10 October 2024).

**Holotype:** KIZR00144, adult male, collected on 28 July 2012 by Dingqi Rao from Langdao Village, Menglian Dai, Lahu, and Wa Autonomous County, Yunnan, China (N22.449°, E99.728° at an elevation of 1158 m a.s.l.).

**Figure 7 animals-14-03030-f007:**
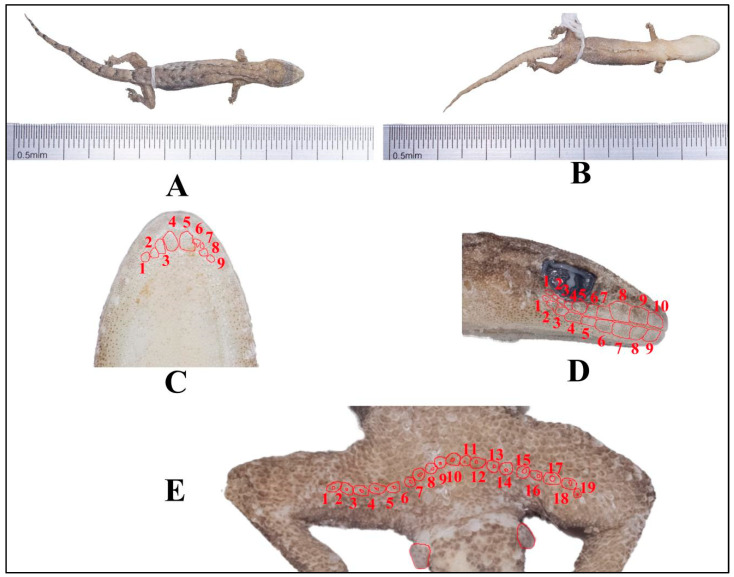
Holotype (KIZR0144) of *Hemiphyllodactylus menglianensis* sp. nov. displaying (**A**) the dorsal view (**B**), ventral view (**C**), and ventral view of the head; red lines indicate chin scales (**D**). In the lateral view of the head, the red lines indicate SL and IL, respectively. (**E**) In the ventral view, red lines indicate precloacal and femoral pores.

**Paratype:** Four adult females (KIZR0085, KIZR0086, KIZR0143 and KIZR2022299), three adult male (KIZR0079, KIZR2022300, KIZR2022301) all from the same locality as the holotype.

**Additional material:** one destroyed specimen from the same locality as the holotype.

**Etymology:** The scientific name “*menglianensis”* is derived from its type locality Menglian County, we suggest Menglian Slender Gecko in English and “孟连半叶趾虎 (Mèng Lián Bàn Yè Zhǐ Hǔ)” in Chinese.

**Diagnosis:** *Hemiphyllodactylus menglianensis* sp. nov. can be distinguished from all other known congeners by a combination of the following characters: maximum SVL of 41.52 mm; 8–10 chin scales; enlarged postmentals; 4 or 5 circumnasal scales; 2–3 internasal scales; 9–11 supralabial scales; 8–10 infralabial scales; dorsal scales 16–18; ventral scales 7–10; a manual lamellar formula of 4–4(5)–(4–6)–4 and a pedal lamellar formula of 4(5)–5–5(6)–4; 16–20 precloacal and femoral pore–bearing scales contiguous in males; 1 cloacal spur on each side; the color of the back of the body is brown; 2 lines of dark blotches on dorsal side running from the neck to sacrum and merge into one; a dark stripe extending from the snout end through the eyes at least to base of neck; dark dorsal transverse blotches and dark postsacral mark bearing anteriorly projecting arms.

**Description of holotype**: Adult male, one longitudinal incision on ventral surface used for liver tissue sampling, small in size (SVL 33.34 mm), and flattened in body shape; head is triangular and elongated (HL/SVL = 0.28), and the dorsum of head is covered in granular scales, which are relatively small; 5 scales surrounding the nostril, including the rostral, the first supralabial and the supranasal scales; three internasal scales, arranged in an isosceles triangle shape; circular mental scale; nine chin scales touching the internal edges of the infralabials, extending from the juncture of the 2nd and 3rd infralabial scales on the left of the mental scale to the same juncture on the right (Chin); scales in the gular region are rounded, non–overlapping, becoming larger and more ovoid on the venter; snout short and narrow (SnW = 1.38 mm; SnW/HL = 0.15); small eyes (ED = 2.00 mm; ED/HL = 0.21); robust body shape (TrunkL/SVL = 0.50); granular scales on the dorsum, with 16 scales within one eye diameter; ventral scales are flattened, with 10 scales within one eye diameter; granular scales on the limbs; Finger I is vestigial, clawless, and with rectangular subdigital lamellae, while Fingers II–V are well–developed; the proximal subdigital lamellae are undivided and rectangular, while the distal subdigital lamellae are divided, angular, and U–shaped, except for the terminal lamellae, which are rounded and undivided; the subdigital lamellae count is indistinguishable; femoral pores and precloacal pores are continuous, with a total count 19, with a single white precloacal pore present on each side. The tail is long (TL/SVL = 0.86), with dorsal scales larger than those on the body and head and smaller than the subcaudals; subcaudals are large and flat.

**Distirbution:** *Hemiphyllodactylus menglianensis* sp. nov. is only known from the type locality in Langdao Village, Menglian Dai, Lahu, and Wa Autonomous County (Figure 1).

**Comparisons:** Appendix A provides a complete comparison of the morphological features of *Hemiphyllodactylus menglianensis* sp. nov. with the *Hemiphyllodactylus* species of clade 3.

*H. menglianensis* sp. nov. differences from *H. mengsongcunensis* sp. nov. by more ventral scales contained with one eye diameter (VS = 7–10 versus 6–8); more dorsal scales contained with one eye diameter (DS = 16–18 versus 11–15); few femoral and precloacal pores (16–20 versus 26–30); dark ventrolateral stripe on trunk absent (versus absent or present).

*H. menglianensis* sp. nov. differences from *H. simaoensis* by longer head (HL/SVL = 0.26–0.28 versus 0.16–0.20); thinner head (HW/HL = 0.59–0.68 versus 0.98–1.18); shorted SnEye distance (SnEye/HL = 0.36–0.42 versus 0.52–0.63); shorted NarEye distance (NarEye/HL = 0.21–0.28 versus 0.38–0.46); smaller eyes (ED/HL = 0.20–0.25 versus 0.30–0.35); narrower snout (SnW/HL = 0.12–0.19 versus 0.18–0.24); few circumnasal scales (CN = 4 or 5 versus 5 or 6); few supralabial scales (SL = 9–11 versus 8–12); few infralabial scales (IL = 8–10 versus 8–11); more ventral scales contained with one eye diameter (VS = 7–10 versus 5–7); more dorsal scales contained with one eye diameter (DS = 16–18 versus 11–15); dark ventrolateral stripe on trunk present (versus absent or indistinct).

*H. menglianensis* sp. nov. differences from *H. jinpingensis* by longer head (HL/SVL = 0.26–0.28 versus 0.17–0.19); thinner head (HW/HL = 0.59–0.68 versus 0.99–1.15); shorted SnEye distance (SnEye/HL = 0.36–0.42 versus 0.51–0.61); shorted NarEye distance (NarEye/HL = 0.21–0.28 versus 0.39–0.45); smaller eyes (ED/HL = 0.20–0.25 versus 0.29–0.38); few circumnasal scales (CN = 4 or 5 versus 5 or 6); few supralabial scales (SL = 9–11 versus 8–12); few infralabial scales (IL = 8–10 versus 9–12); more ventral scales contained with one eye diameter (VS = 7–10 versus 5–9); more dorsal scales contained with one eye diameter (DS = 16–18 versus 10–15); few femoral and precloacal pores (16–20 versus 20–24); few clocal spur (1 versus 1 or 2); dark dorsolateral stripe on trunk absent (vs. present).

**Natural History:** All specimens were collected in Langdao Village, Menglian Dai, Lahu, and Wa Autonomous County. These specimens are found on both the exterior and interior walls of the residential area, and we can see them preying on mosquitoes.

**Variation:** the alterations in the morphology and coloration of *Hemiphyllodactylus menglianensis* sp. nov. are detailed in Appendix A.

**Coloration in ethanol:** except for KIZR0086, which has a gray–white body, the heads, bodies, and dorsal surfaces of the limbs of the other individuals are brown. There is at least one stripe in the preorbital and postorbital regions that extend to the neck. The ventral portion of the head and anterior half of the body is beige, transitioning to dark brown on the posterior half. Some light black spots are present on the dorsal surfaces of the limbs, while the ventral surfaces are beige. The tail tips of regenerated tails lack patterns, appearing uniformly dark brown, while the colors of the intact tail are consistent with the body color but slightly lighter.

### 4.2. Hemiphyllodactylus mengsongcunensis sp. nov. (Figure 8)

https://zoobank.org/urn:lsid:zoobank.org:act:90118672-F0C0-42F5-A0DB-C47825B2B4B3 (accessed on 10 October 2024).

**Holotype:** KIZ2023576, adult female, collected on 26 August 2023 by Ziqi Shen, Chaoying Yuan, Yvxin Fan and Shuangshuang Wu from Mengsong Village, Jinghong, Yunnan, China (N21.492°, E100.510° at an elevation of 1594 m a.s.l.).

**Paratype:** one adult male (KIZR00106) collected on May 2012 by Dingqi Rao; three adult males (KIZR2023578, KIZR2023583, KIZR2023606) collected on 26 August 2023 by Ziqi Shen, Chaoying Yuan, Yvxin Fan and Shuangshuang Wu from Mengsong Village, Jinghong; three adult females (KIZR00110, KIZR00125, KIZR00126) collected on May 2012 by Dingqi Rao; four adult female (KIZR2023571, KIZR2023577, KIZR2023589, KIZR2023590) collected on 26 August 2023 by Ziqi Shen, Chaoying Yuan, Yvxin Fan and Shuangshuang Wu from Mengsong Village, Jinghong and one destroyed specimens collected on May 2012 by Dingqi Rao, all from the same locality as the holotype.

**Etymology:** The scientific name “*mengsongcunensis”* is derived from its type locality Mengsong Village, Jinghong City, we suggest Mengsongcun Slender Gecko in English and “勐宋村半叶趾虎 (Měng Sòng cūn Bàn Yè Zhǐ Hǔ)” in Chinese.

**Diagnosis:** *Hemiphyllodactylus mengsongcunensis* sp. nov. can be distinguished from all other known congeners by a combination of the following characters: maximum SVL of 45.6 mm; 6–10 chin scales; enlarged postmentals; 5 circumnasal scales; 2–3 internasal scales; 8–11 supralabial scales; 8–11 infralabial scales; dorsal scales 11–15; ventral scales 6–8; a manual lamellar formula of 3(4)–(4–6)–(4–7)–4(5) and a pedal lamellar formula of 4(5)–(4–6)–(4–6)–4(5); 26–30 precloacal and femoral pore-bearing scales contiguous in males; 1 or 2 cloacal spurs on each side; dark postorbital stripes; the color of the back of the body is light gray; two lines of dark blotches running from neck to sacrum on dorsal side; ventrolateral stripe on trunk present or absent and a dark postsacral mark bearing anteriorly projecting arms.

**Description of holotype:** Adult female, with a longitudinal incision on the ventral surface used for liver tissue sampling, larger in size (SVL 45.6 mm), and somewhat flattened in body shape; head triangular, elongated (HL/SVL = 0.26); dorsum of head covered in granular scales, which are relatively small; five supralabials, with the lower two being the rostral and the largest upper labial, while the other three are circular; three internasal scales, arranged in an isosceles triangle shape; circular mental scale; eight chin scales touching the internal edges of the infralabials, extending from the juncture of the 2nd and 3rd infralabial scales on the left of the mental scale to the same juncture on the right (Chin); nostril scale divided on both sides, with four scales on each side; scales in the gular region are rounded, non–overlapping, becoming larger and more ovoid on the venter, Short and narrow snout (SnW = 1.64 mm; SnW/HL = 0.14); small eyes (ED = 2.72 mm; ED/NarEye = 0.77); robust body shape (TrunkL/SVL = 0.55); granular scales on the dorsum, with 15 scales within one eye diameter; ventral scales are flattened, with 7 scales within one eye diameter; granular scales on the limbs; Finger I is vestigial, clawless, and with rectangular subdigital lamellae, while Fingers II–V are well-developed; the proximal subdigital lamellae are undivided and rectangular, while the distal subdigital lamellae are divided, angular, U–shaped, except for the terminal lamellae, which are rounded and undivided; the forefoot and hindfoot have a digital formulae of 4–4–6(7)–4(5) and 4–4–5–4 respectively; femoral pores and precloacal pores are absent, with a single white precloacal pore present on each side. Tail length (TL/SVL = 1.04), with dorsal scales on the tail larger than those on the body and head, but smaller than the subcaudals. The ventral scales are large and flat.

**Figure 8 animals-14-03030-f008:**
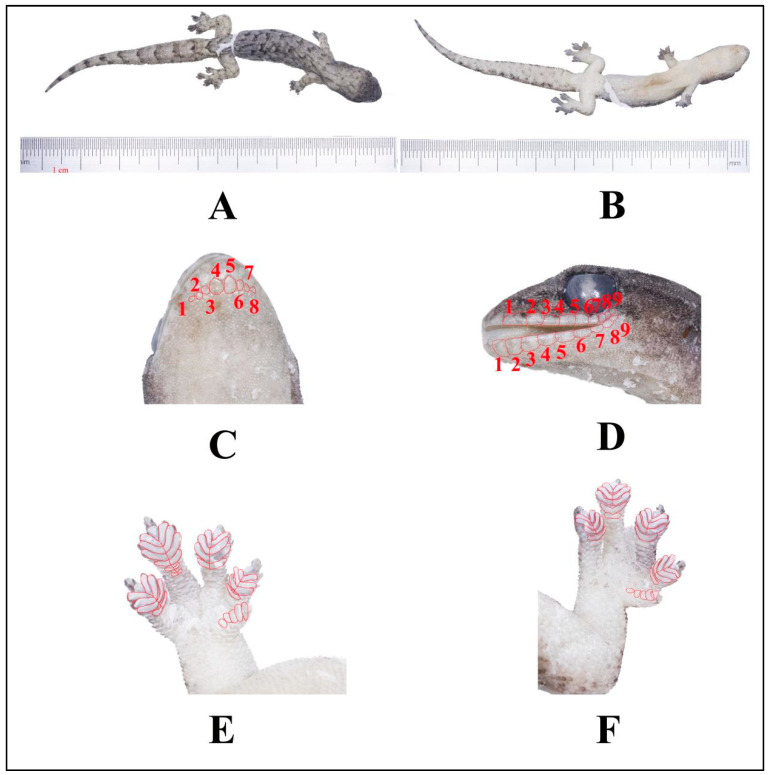
Holotype (KIZR2023576) of *Hemiphyllodactylus mengsongcunensis* sp. nov. displaying (**A**) the dorsal view, (**B**) the ventral view, and (**C**) the ventral view of the head; red lines indicate chin scales; (**D**) in lateral view of the head, red lines indicate SL and IL, respectively; (**E**,**F**) in view of lamellae formula counting on fingers and feet I to V.

**Distirbution:** *Hemiphyllodactylus mengsongcunensis* sp. nov. is only known from the type locality in Mengsong Village, Jinghong, Yunnan, China (Figure 1).

**Comparisons:** Appendix A provides a complete comparison of the morphological features of *H. mengsongcunensis* sp. nov. with the *Hemiphyllodactylus* species of clade 3.

The difference between *H. mengsongcunensis* sp. nov. and *H. menglianensis* sp. nov. has already been described above.

*H. mengsongcunensis* sp. nov. differs from *H. simaoensis* by its longer head (HL/SVL = 0.25–0.29 versus 0.16–0.20); thinner head (HW/HL = 0.63–0.77 versus 0.98–1.18); shorted SnEye distance (SnEye/HL = 0.37–0.47 versus 0.52–0.63); shorted NarEye distance (NarEye/HL = 0.26–0.32 versus 0.38–0.46); smaller eyes (ED/HL = 0.19–0.24 versus 0.30–0.35); narrower snout (SnW = 0.12–0.15 versus 0.18–0.24); few circumnasal scales (CN = 5 versus 5 or 6); few internasal scales (IS = 2 or 3 versus 1–4); few supralabial scales (SL = 8–11 versus 8–12); more ventral scales contained with one eye diameter (VS = 6–8 versus 5–7); more femoral and precloacal pores (26–30 versus 16–27); more cloacal spur (1 or 2 versus 1); dark dorsolateral stripe on trunk absent (versus indistinct); and dark ventrolateral stripe on trunk absent or present (versus absent).

*H. mengsongcunensis* sp. nov. differ from *H. jingpingensis* by its longer head (HL/SVL = 0.25–0.29 versus 0.17–0.19); thinner head (HW/HL = 0.63–0.77 versus 0.99–1.15); shorted SnEye distance (SnEye/HL = 0.37–0.47 versus 0.51–0.61); shorted NarEye distance (NarEye/HL = 0.26–0.32 versus 0.39–0.45); smaller eyes (ED/HL = 0.19–0.24 versus 0.29–0.38); narrower snout (SnW = 0.12–0.15 versus 0.18–0.24); few chin scales (Chin = 6–10 versus 7–10); few circumnasal scales (CN = 5 versus 5 or 6); few internasal scales (IS = 2 or 3 versus 1–5); few supralabial scales (SL = 8–11 versus 8–12); few infralabial scales (IL = 8–11 versus 9–12); more femoral and precloacal pores (26–30 versus 20–24); dark dorsolateral stripe on trunk absent (versus present); dark ventrolateral stripe on trunk absent or present (versus absent).

**Natural History:** All specimens were collected on the walls of abandoned houses in Mengsong Village and roadside walls with small gaps and holes. When they were startled, their entire bodies curled up in the gaps.

**Variation:** the alterations in the morphology and coloration of *Hemiphyllodactylus mengsongcunensis* sp. nov. are detailed in Appendix A.

**Coloration in ethanol.** The coloration of the ventral side of the head is dark gray, distinct from the rest of the head. With the exception of the individual KIZR202300583, which has a dark color, the ventral color of the body and limbs of the other specimens is gray, with some individuals having slightly lighter ventral color on the limbs. The preorbital and postorbital stripes extend to the neck, but in some individuals, the preorbital stripe is lighter. The dorsal surface is covered with large, scattered, or regular black spots. The color of the ventral surface of the head and body is light gray, while in a few individuals, it transitions to dark gray on the ventral surface. The dorsal surface of the limbs has scattered black spots, while the ventral surface is light gray. The tail tip is dark. The ventral surface of the intact tail is reddish–orange or light gray, while the regenerated tail is uniformly gray. The femoral pores are white or matching with the color of the skin.

### 4.3. Hemiphyllodactylus jinghongensis sp. nov. (Figure 9)

https://zoobank.org/urn:lsid:zoobank.org:act:E1BF0443-4231-42EE-A5DA-3CA7A0289108 (accessed on 10 October 2024).

**Figure 9 animals-14-03030-f009:**
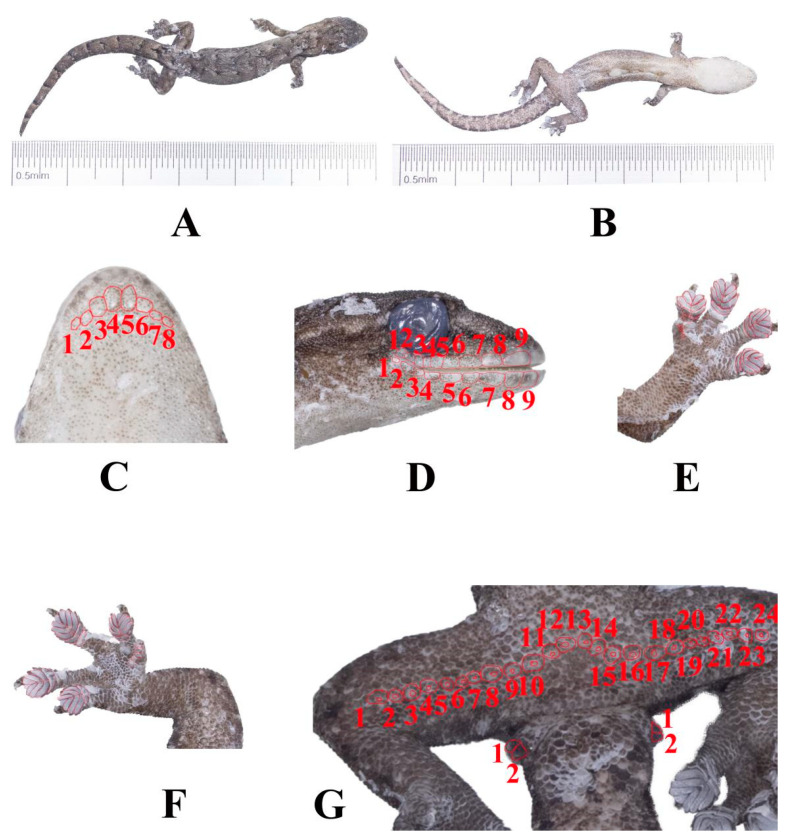
Holotype (KIZR2023579) of *Hemiphyllodactylus jinghongensis* sp. nov. displaying (**A**) the dorsal view (**B**), the ventral view, and (**C**) the ventral view of the head; red lines indicate chin scales. (**D**) In the lateral view of the head, red lines indicate SL and IL, respectively. (**E**,**F**) View of lamellae formula counting on fingers and feet I to V; (**G**) in ventral view, red lines indicate femoral and precloacal pores and cloacal spurs.

**Holotype:** KIZR2023579, adult male, collected on 26 August 2023 by Ziqi Shen, Chaoying Yuan, Yvxin Fan and Shuangshuang Wu from Mengsong Village, Jinghong, Yunnan, China (N21.492°, E100.510° at an elevation of 1594 m a.s.l.).

**Paratype:** One adult female (KIZR00124) collected on May 2012 by Dingqi Rao; six adult female (KIZR2023580, KIZR2023588, KIZR2023601, KIZR2023605, KIZR2023607 and KIZR2023613) collected on 26 August 2023 by Ziqi Shen, Chaoying Yuan, Yvxin Fan and Shuangshuang Wu from Mengsong Village, Jinghong; three adult males (KIZR0092, KIZR00103 and KIZR00111) collected on May 2012 by Dingqi Rao, all from the same locality as the holotype.

**Etymology:** The scientific name “*jinghongensis”* is derived from its type locality Jinghong City, Mengsong Village, Jinghong and we suggest Jinghong Slender Gecko in English and “景洪半叶趾虎 (Jǐng Hóng Bàn Yè Zhǐ Hǔ)” in Chinese.

**Diagnosis:** *Hemiphyllodactylus jinghongensis* sp. nov. can be distinguished from all other known congeners by a combination of the following characters: maximum SVL of 42.66 mm; 7–10 chin scales; enlarged postmentals; 5 circumnasal scales; 1–4 internasal scales; 8–11 supralabial scales; 8–10 infralabial scales; dorsal scales 12–16; ventral scales 7–9; a manual lamellar formula of 3(4)–4(5)–(4–6)–4(5) and a pedal lamellar formula of 4–4(5)–(4–6)–4(5); 22–24 precloacal and femoral pore–bearing scales contiguous in males; 1 cloacal spur on each side; the color of the back of the body is grayish brown; dark postorbital stripes; dorsolateral light-colored spots on trunk present; two lines of bicolor transverse blotches running from neck to sacrum on dorsal side; ventrolateral stripe on trunk present; a dark postsacral mark bearing anteriorly projecting arms and transverse spots on the back of the tail present, which are dark and light brown in color.

**Description of holotype:** Adult male, with a longitudinal incision on the ventral surface used for liver tissue sampling, small in size (SVL = 38.44 mm), and somewhat flattened in body shape; head triangular, elongated (HL/SVL = 0.25); dorsum of head covered in granular scales, which are relatively small; 5 supralabials, with the lower two being the rostral and the largest upper labial, while the other three are circular; 3 internasal scales, arranged in an isosceles triangle shape; circular mental scale; 8 chin scales touching the internal edges of the infralabials, extending from the juncture of the 2nd and 3rd infralabial scales on the left of the mental scale to the same juncture on the right (Chin); nostril scale divided on both sides, with 4 scales on each side; scales in the gular region are rounded, non-overlapping, becoming larger and more ovoid on the venter, Short and narrow snout (SnW = 1.38 mm; SnW/HL = 0.19); small eyes (ED = 2.32 mm; ED/HL = 0.24); robust body shape (TrunkL/SVL = 0.55); granular scales on the dorsum, with 14 scales within one eye diameter; ventral scales are flattened, with 8 scales within one eye diameter; granular scales on the limbs; Finger I is vestigial, clawless, and with rectangular subdigital lamellae, while Fingers II–V are well–developed; the proximal subdigital lamellae are undivided and rectangular, while the distal subdigital lamellae are divided, angular, and U-shaped, except for the terminal lamellae, which are rounded and undivided; the forefoot and hindfoot have a digital formulae of 4–4–4–4 and 4–5–4–4 respectively; continuous femoral and precloacal pores, numbering 24; 1 white cloacal spur present on each side. Tail length (TL/SVL = 0.89), with dorsal scales on the tail larger than those on the body and head, but smaller than the subcaudals. The ventral scales are large and flat.

**Distirbution:** *Hemiphyllodactylus jinghongensis* sp. nov. is only known from the type locality in Mengsong Village, Jinghong, Yunnan, China (Figure 1).

**Comparisons:** Appendix A provides a complete comparison of the morphological features of *Hemiphyllodactylus jinghongensis* sp. nov. with *H. zhutangxiangensis*, *H. gengmaensis*, and *H. changningensis*.

*H. jinghongensis* sp. nov. differences from *H. zhutagnxiangensis* by longer head (HL/SVL = 0.24–0.28 versus 0.17–0.20); smaller SnEye distance (SnEye/HL = 0.39–0.44 versus 0.53–0.60); smaller NarEye distance (NarEye/HL = 0.26–0.34 versus 0.39–0.44); smaller eyes (ED/HL = 0.22–0.25 versus 0.30–0.36); narrower snout (SnW/HL = 0.13–0.16 versus 0.19–0.22); more chin scales (Chin = 7–10 versus 7–9); more ventral scales contained with one eye diameter (VS = 7–9 versus 5–7); more dorsal scales contained with one eye diameter (DS = 12–16 versus 11–15); more femoral and precloacal pores in males (22–24 versus 20–23); dorsolateral light–colored spots on trunk present (versus absent); dark ventrolateral stripe on trunk present (versus absent).

*H. jinghongensis* sp. nov. differs from *H. gengmaensis* by its shorter SnEye distance (SnEye/HL = 0.36–0.41 versus 0.39–0.44); greater NarEye distance (NarEye/HL = 0.26–0.34 versus 0.24–0.30); narrower snout (SnW/HL = 0.13–0.16 versus 0.15–0.24); more chin scales (Chin = 7–10 versus 8–9); fewer circumnasal scales (CN = 5 versus 6); more internasal scales (IS = 1–4 versus 2–3); more infralabial scales (IL = 8–10 versus 8–9); fewer ventral scales contained with one eye diameter (VS = 7–9 versus 8–10); fewer subdigital lamellae wider than long on first toe(SL1T = 5 versus 6); dorsolateral light-colored spots on trunk present (versus absent); dark ventrolateral stripe on trunk present (versus absent) and dark reticulate pattern on dorsum absent (versus present or indistinct).

*H. jinghongensis* sp. nov. differs from *H. changningensis* by longer trunk (TrunkL/SVL = 0.48–0.54 versus 0.46–0.51); narrower head (HW/HL = 0.63–0.73 versus 0.72–0.80); shorted SnEye distance (SnEye/HL = 0.39–0.44 versus 0.41–0.49); shorted NarEye distance (NarEye/HL = 0.26–0.34 versus 0.30–0.37); great eyes (0.68–0.89 versus 0.61–0.77); wider snout (SnW/HL = 0.19–0.24 versus 0.16–0.20); more chin scales (Chin = 7–10 versus 7 or 8); more circumnasal scales (CN = 5 versus 3 or 4); more internasal scales (IS = 1–4 versus 2 or 3); more ventral scales contained with one eye diameter (VS = 7–9 versus 6–8); more dorsal scales contained with one eye diameter (DS = 12–16 versus 11–15); more subdigital lamellae wider than long on first finger and toe(SL1T = 5 versus 3 or 4); more femoral and precloacal pores in males (22–24 versus 19–22); dorsolateral light–colored spots on trunk present (versus absent); dark ventrolateral stripe on trunk present (versus absent). dark reticulate pattern on dorsum absent (versus present); postsacral marking anteriorly projecting arms present (versus absent).

**Natural History:** All specimens were collected on the walls of abandoned houses in Mengsong Village (Figure 10) and roadside walls with small gaps and holes. When they were startled, their entire bodies curled up in the gaps.

**Variation:** the alterations in the morphology and coloration of *Hemiphyllodactylus jinhongensis* sp. nov. are detailed in Appendix A.

**Coloration in ethanol:** the heads, bodies, and dorsal surfaces of the limbs are all dark gray, with preorbital and postorbital markings present. In some individuals, the postorbital marking extends to the base of the legs, forming lateral markings on the ventral surface. Some individuals have no markings on the dorsal side, appearing uniformly dark gray, while others display black spots with white dots on the back. The ventral side of the head is grayish white, differing from the body, while the ventral surface of the body is a gradient of gray with dark spots. The dorsal surfaces of the limbs have scattered small black spots. The tail tips are dark, with the intact ventral tail being grayish white, and the regenerated tail being gray without patterns. Finally, the femoral pores are white.

## 5. Discussion

Our PCA and DAPC results indicate that the newly identified species exhibits minimal overlap with the previously published species, which is not an isolated case within this taxon. Morphological overlap has been observed among published species, such as between *H. jinpingensis* and *H. simaoensis*, as well as between *H. simaoensis* and *H. ngwelwini* [9]. Consequently, we concur with the perspective of Grismer et al. (2013) [11] that this taxon comprises a morphologically conservative group, necessitating the application of integrative taxonomic approaches to enhance the accuracy of species identification within this group.

Our research has essentially clarified the taxonomic status of the *Hemiphyllodactylus* species in the southern and western parts of Yunnan. Based on morphological and genetic data, we have described three new species. *Hemiphyllodactylus jinghongensis* sp. nov. is divided into clade 3. All species of clade 3, excluding *H. zalonicus*, being found in the western and southern regions of Yunnan Province, China. The description of *H. jinhongensis* has extended the boundary of species in clade 3, 152 km to the south. This implies the potential existence of other cryptic species in the easternmost part of Myanmar and the northern region of Thailand. On the other hand, *H. menglianensis* sp. nov. and *H. mengsongcunensis* sp. nov. are both members of clade 4, which are distributed in Myanmar and China. Previously, there was an approximately 500 km biogeographic gap between different members of clade 4, separated by the Shan Plateau in Myanmar, due to the political and military complexities in northern Myanmar, research on amphibians and reptiles in the Shan Plateau is severely limited [17].

Grismer et al. (2015) [16] demonstrated instances of repeated evolution in *Hemiphyllodactylus* on the Malaysian sky islands. Our research shows that two species of *Hemiphyllodactylus* can be found in sympatry in Jinghong City, Nanhua County and Lvchun County in Yunnan Province, China, suggesting that this is not a unique event; it is possible that this event is widespread within this genus.

Currently, biogeographical studies on *Hemiphyllodactylus* are confined to specific regions and a limited number of species [11,21]. No comprehensive published study exists on their historical colonization and distribution patterns of *Hemiphyllodactylus*. This is partly due to the shortage of specimen collection sites, as well as the complexity of geological events in the Southeast Asian region. Our new discovery shows that *Hemiphyllodactylus* in the karst regions of Yunnan has had a complex colonization history, given that we potentially found multiple species in one site (*H. mengsongcunensis* sp. nov. and *H. jinghongensis* sp. nov.) as well as one species distributed in multiple karst hills (*H. jinpingensis*, *H. changningensis*). Moreover, our study shows *H. simaoensis* has spread from Ninger County to Shangyong Village in Mengla County, but Agung et al. (2022) [9] reported that a specimen from Mengyuan Town in Mengla County (Mengyuan Town is located between Ninger County and Shangyong Village) belongs to *H. yunnanensis* complex. This suggests the possibility of unkonwn species belonging to *H. yunnanensis* complex and additional colonization routes in northern Laos and Vietnam. Given the wide distribution range of *Hemiphyllodactylus* species, resolving the biogeographical issues of the entire genus necessitates collecting specimens from multiple locations. Therefore, we hope that further international cooperation can address this issue. Additionally, our study identified several distribution locations of the *H. yunnanensis* complex, including Chuxiong City and Nanhua County.

## 6. Conclusions

Our research has clarified the taxonomic status of *Hemiphyllodactylus* species in the unexplored southern and western parts of Yunnan, providing crucial foundational data for their taxonomy and biogeography. Moreover, three new species of the genus *Hemiphyllodactylus* have been described: *H. menglianensis* sp. nov., *H. mengsongcunensis* sp. nov. and *H. jinghongensis* sp. nov. based on the analysis of molecular and morphological data. Moreover, the co-distribution of multiple species across various locations and four discoveries of new OTUs (three know and one unknown) demonstrate the high species diversity of the genus *Hemiphyllodactylus* within Yunnan Province. Therefore, it is possible that there are still numerous undiscovered cryptic species in Yunnan Province.

## Figures and Tables

**Figure 1 animals-14-03030-f001:**
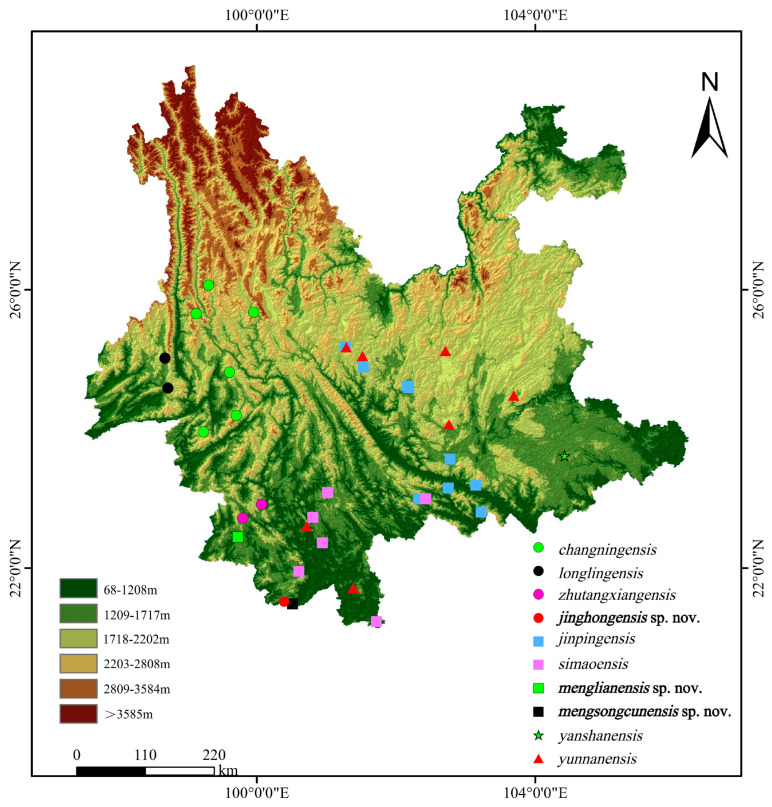
Distribution map of the genus *Hemiphyllodactylus* in Yunnan Province, China. The circle represents the species of clade 3, the square represents the species of clade 4, the star represents the species of clade 6, and the triangle represents the *Hemiphyllodactylus yunnanensis* complex.

**Figure 2 animals-14-03030-f002:**
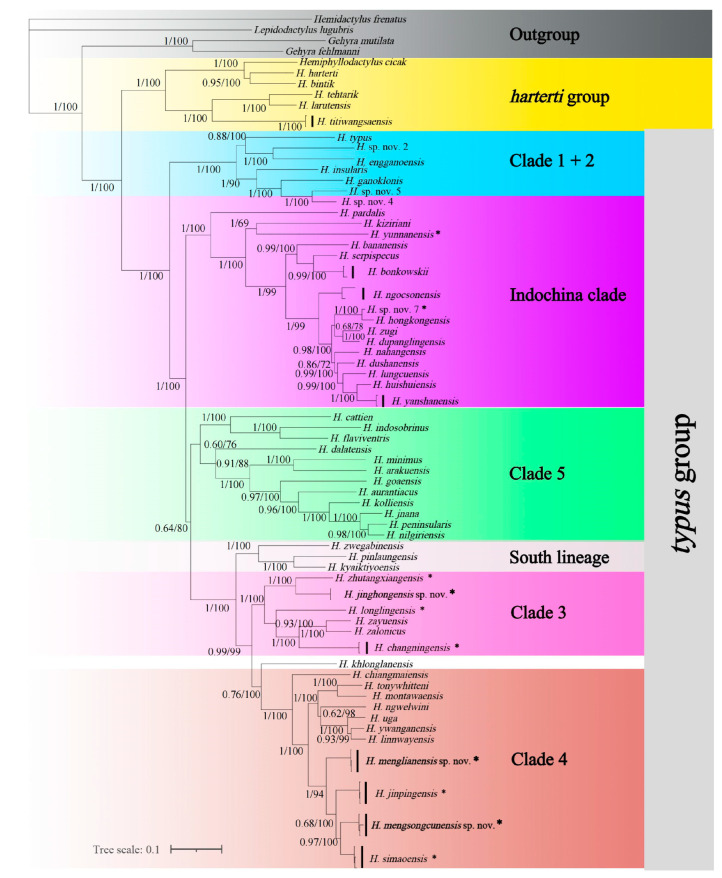
Maximum likelihood consensus tree of *Hemiphyllodactylus* with Bayesian posterior probabilities (BPP) and Ultrafast Bootstrap support (UFB) value coding at the nodes. Group nomenclature follows Agung et al. (2021) [8]. * indicates those specimens from this study.

**Figure 3 animals-14-03030-f003:**
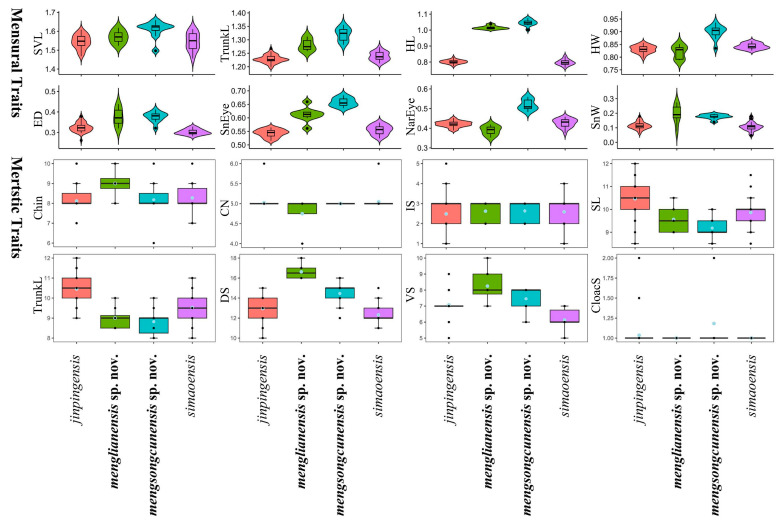
Differences in adjusted mensural (**upper**) and meristic (**below**) traits between *H. menglianensis* sp. nov., *H. mengsongcunensis* sp. nov. and their related species.

**Figure 4 animals-14-03030-f004:**
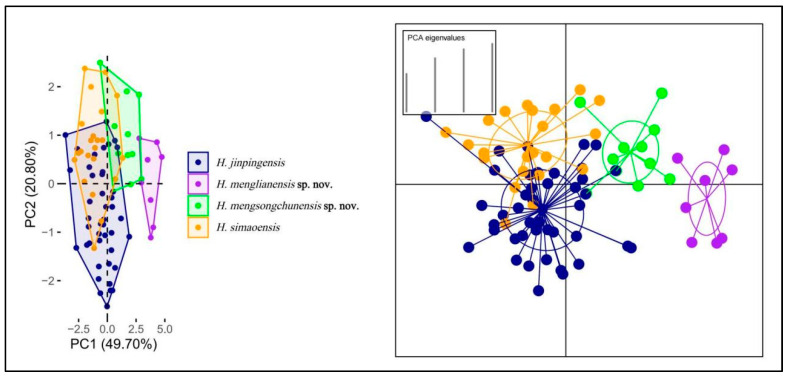
PCA (**left**) of *Hemiphyllodactylus* species between new species with their related species showing their morphospatial relationships along the first two principal components and DAPC (**right**) based on retention of the first five PCs.

**Figure 5 animals-14-03030-f005:**
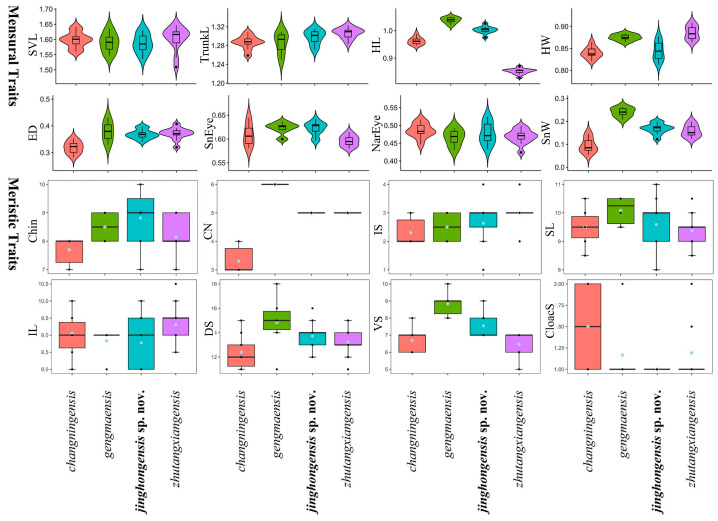
Differences in adjusted mensural (**upper**) and meristic (**below**) traits between *H. jinghongensis* sp. nov. and its closely related species.

**Figure 6 animals-14-03030-f006:**
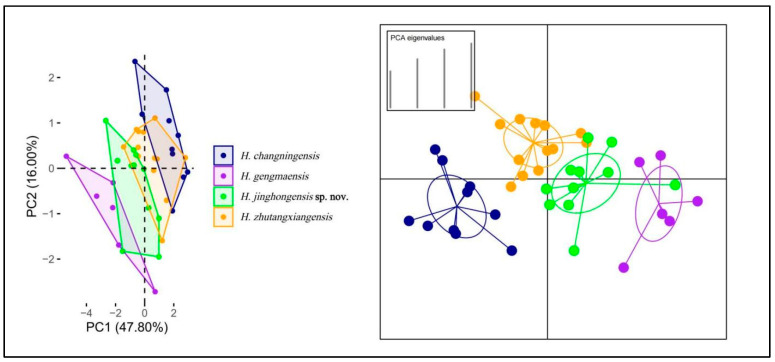
PCA (**left**) of *Hemiphyllodactylus* species between new species with their related species showing their morphospatial relationships along the first two principal components and DAPC (**right**) based on retention of the first five PCs.

**Figure 10 animals-14-03030-f010:**
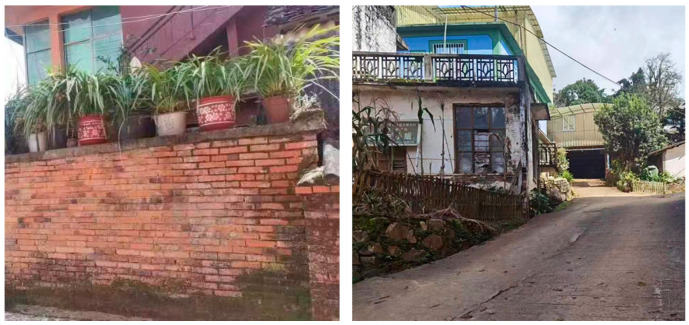
Site of discovery for *Hemiphyllodactylus mengsongcunensis* sp. nov. and *Hemiphyllodactylus jinghongensis* sp. nov.

**Table 1 animals-14-03030-t001:** Location of the 17 selected karst field sites.

Species Name (No of Specimens)	Location	Longitude (°)	Latitude (°)	Altitude (m, a.s.l.)
*mengsongcunensis* sp. nov. & *jinhongensis* sp. nov. (24)	Mengsong village, Jinghong City	21.492	100.510	1594
*menglianensis* sp. nov. (9)	Langdao town, Menglian County	22.449	99.728	1158
*zhutangxiangensis* (1)	Donghe town, Lancang County	22.922	100.069	1871
*longlingensis* (2)	Xinzhai village, Tengchong City	25.020	98.681	1229
*changningensis* (9)	Jiancao village, Yunlong County	26.068	99.308	1945
*changningensis* (5)	Caojian town, Yunlong County	25.660	99.127	2067
*changningensis* (1)	Wumulong village, Yongde County	24.197	99.705	1935
*changningensis* (1)	Shuiping village, Yangbi County	25.679	99.955	1640
*simaoensis* (3)	Dadugang village, Jinghong City	22.366	100.941	1341
*simaoensis* (1)	Shangyong village, Mengla County	21.237	101.717	752
*simaoensis* (1)	Nannuoshan village, Menghai County	21.955	100.603	1324
*simaoensis and jinpingensis* (21)	Lvchun County	22.996	102.422	1613
*jinpingensis* (6)	Ziwu village, Chuxiong City	24.890	101.531	1875
*jinpingensis* (14)	Xinjie Town, Yuanyang County	23.153	102.748	1604
*jinpingensis* (4)	Shang village, Yimen County	24.597	102.180	1667
*jinpingensis and yunnanesis* (8)	Nanhua County	25.182	101.282	1846
*yunnanensis* (3)	Chuxiong City	25.053	101.516	1780
*H*. sp. (1)	N/A	N/A	N/A	N/A

**Table 2 animals-14-03030-t002:** The mean percentage of the uncorrected *p*–distance among the clade 3 of *Hemiphyllodactylus* species studied based on mitochondrial ND2 gene fragments.

Species Name	1	2	3	4	5	6
1. *H. longlingensi*	–					
2. *H. zalonicus*	16.1	–				
3. *H. zhutangxiangensis*	21.9	16.6	–			
4. *H. changningensis*	23.3	14.5	22.2	–		
5. *H. gengmaensis*	19.8	11.5	19.1	9.7	–	
**6. *H. jinghongensis* sp. nov.**	**21.0**	**15.5**	**8.5**	**18.6**	**17.1**	**0.2**

**Table 3 animals-14-03030-t003:** The mean percentage of the uncorrected *p*–distance among the clade 4 of *Hemiphyllodactylus* species studied based on mitochondrial ND2 gene fragments.

Species Name	1	2	3	4	5	6	7	8	9	10	11
1. *H. linnwayensis*	–										
2. *H. montawaensis*	8.9	–									
3. *H. ngwelwini*	7.3	9.1	–								
4. *H. simaoensis*	9.3	9.3	9.6	–							
5. *H. chiangmaiensis*	12.3	11.3	11.6	11.3	–						
6. *H. tonywhitteni*	7.4	4.4	7.9	9.1	10.1	–					
7. *H. uga*	4.0	8.5	8.1	10.2	10.6	6.6	–				
8. *H. jinpingensis*	10.1	10.9	10.7	4.9	12.6	10.6	11.4	–			
9. *H. ywanganensis*	2.2	8.6	7.4	9.4	11.4	6.3	3.0	10.2	–		
**10. *H. menglianensis* sp. nov.**	**7.6**	**8.7**	**9.2**	**5.7**	**11.1**	**7.7**	**8.7**	**6.0**	**7.7**	**0.3**	
**11. *H.mengsongchunensis* sp. nov.**	**12.6**	**12.9**	**13.0**	**5.2**	**16.4**	**12.6**	**13.1**	**7.5**	**12.8**	**8.2**	**0.3**

**Table 4 animals-14-03030-t004:** Significant *p*–values from the results of the ANOVA and Turkey HSD analyses comparing all combinations of species pairs between new species with their related species in clade 4.

Characters	SVL	TrunkL	HL	HW	ED	SnEye	NarEye	SnW	CN	SL	IL	DS	VS
*H. jinpingensis* vs. *H. menglianensis* sp. nov.		<0.0001	0		0.01	0	0.0003		0.006	0.01	<0.0001	0	0.0001
*H. jinpingensis* vs. *H. mengsongchunensis* sp. nov.	<0.0001	0	0	<0.0001	0.001	0	0	<0.0001		<0.0001	<0.0001	0.0006	
*H. menglianensis* sp. nov. vs. *H. mengsongchunensis* sp. nov.				<0.0001		0.03				0.01	<0.0001		<0.0001
*H. jinpingensis* vs. *H. simaoensis*		<0.0001	0.0002	0.04	0.001	<0.0001	0					0.0003	
*H. menglianensis* sp. nov. vs. *H. simaoensis*		<0.0001	0		<0.0001	<0.0001	<0.0001		0.007			0	<0.0001
*H. mengsongchunensis* sp. nov. vs. *H. simaoensis*	0.0002	0	0	0.02	<0.0001	0	0	<0.0001			0.04	<0.0001	<0.0001

**Table 5 animals-14-03030-t005:** Significant *p*–values from the results of the ANOVA and Turkey HSD analyses comparing all combinations of species pairs between new species with its related species in clade 3.

Characters	TrunkL	HL	HW	ED	SnEye	SnW	Chin	CN	DS	VS	CloacS
*H. changningensis* vs. *H. gengmaensis*		0	0.02	0.0001		<0.0001		<0.0001	0.01	<0.0001	
*H. changningensis* vs. *H. jinghongensis* sp. nov.		<0.0001		0.0002		<0.0001	0.006	0.0006		0.04	0.03
*H. gengmaensis* vs. *H. jinghongensis* sp. nov.	0.02	0	0.04	<0.0001		<0.0001		0.008			
*H. changningensis* vs. *H. zhutangxiangensis*		<0.0001	0.0002			<0.0001		0.0004		0.004	
*H. gengmaensis* vs. *H. zhutangxiangensis*	0.04	0			0.009	<0.0001		0.009		<0.0001	
*H. jinghongensis* sp. nov. vs. *H. zhutangxiangensis*		0	0.0004		0.002					0.002	

## Data Availability

The original contributions presented in the study are included in the article and Appendix A, further inquiries can be directed to the corresponding authors.

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
