# Peer review of "Integrative Taxonomy Revealed High Diversity of Hemiphyllodactylus Bleeker, 1860 (Squamata: Gekkonidae) and the Description of Three New Species from Yunnan Province, China"

_animals, 2024, doi:10.3390/ani14203030_

Round 1
Reviewer 1 Report
Comments and Suggestions for Authors
This paper describes three new species of gecko from Yunnan region, China based on genetic analyses and well-developed morphological analyses. Overall, I find that the evidence used to describe this new taxa are valid and sufficient, therefore most of my suggestions below are several but minor.
My only word of caution is that the authors did not seem to explore all the evidence that support species delimitations. Hemiphyllodactylus seems to be a group that have been described on the limit to be consider over split in some clades. For that reason, I would like to see analyses that test gene flow between the different already recognized taxa, in this work and future works, exploring nuclear and mitochondrial evolution, separately. It is true, that the authors in this work have enough evidence that support the specific status for their proposed candidate new species, however, the work will be benefit if the author can at least generate nuclear markers (i.e., RAG1) to see if the nuclear marker support such taxonomic decision. Also, I think the species account can be reorder in order to have a more coherent flow. Thus, I would list the sections as follow: Holotype, Paratype, Etymology, Diagnosis, Comparisons, Description of holotype, Variation, Coloration, Distribution and Natural History. Moreover, in all the figures of the specimens I would remove the red lines… they are confusing and difficult the good visualization of the scalation. I list below a list of comments that I hope can help to improve the manuscript. Of course, you can ignore whatever you consider but I think is worth to take it into consideration.
Finally, I think the discussion section need more work. The assumptions and assertation state in this section are not well supported or explained, by their evidence and the references used. Also, I have missed some discussion about the relevance of this area for other groups of reptiles, and how their results can contribute to the knowledge on the herpetological evolution and diversification in the area. Also, the conservation implication of identifying such “micro-endemic” species, since most of their records are in urban areas… are this because they invest most the work in human disturbed areas? Are natural areas surrounding the cities disturbed? Etc… I think there is a great story behind this work that is worth of discussion.
I attached you here a word file with all the detail comments...

Some of the grammar needs improvement- I suspect from non-native English writers (me neither)- and I have endeavored to make a few corrections here and there to help them, but probably needs a more careful proofread.
Reviewer 2 Report
Comments and Suggestions for Authors
Dear authors,
I have completed the review of your manuscript that describes three new species of Hemiphyllodactylus. I agree with the authors that these are new species and the data presented is substantial and supports the authors' claims. However, there are a few issues with the manuscript that must be addressed before it can be published.
I have marked some comments in the attached pdf. The authors may choose to address them to enhance their paper.
Best,

Overall, the manuscript has flow, but the language may be improved a bit.
Reviewer 3 Report
Comments and Suggestions for Authors
The present article is a good writing study with novelty and important results that contribute to the knowledge of biodiversity. The phylogeny Hemiphyllodactylus was elucidated and 3 new species were described with significant node support and morphological characters. I only have minor comments
Specific comments
The simple summary has not been presented
The citation format does not correspond to Animals journal
Include a sentence that describes how the lizards were captured.
Indicate the program that was used to draw the phylogenetic tree
Indicate the program that was used to draw the map
Indicate the substitution model with which the uncorrected pairwise divergence was calculated
Cite Figure 1 before placing it in the manuscript
Line 31. Delete the repeated word “and”
In the tree of figure 2 it would be very good to increase the thickness of the branches, the font size and node numbers.
On the tree, mark the new species in the same form as the one on the map.
Cite table 4 before placing it in the manuscript.
It would be perfect to Include some photographs of the specimens in their natural environment.
